# Chemico-Physical Properties of Some 1,1′-Bis-alkyl-2,2′-hexane-1,6-diyl-bispyridinium Chlorides Hydrogenated and Partially Fluorinated for Gene Delivery

**DOI:** 10.3390/molecules28083585

**Published:** 2023-04-20

**Authors:** Michele Massa, Mirko Rivara, Thelma A. Pertinhez, Carlotta Compari, Gaetano Donofrio, Luigi Cristofolini, Davide Orsi, Valentina Franceschi, Emilia Fisicaro

**Affiliations:** 1Department of Maternal Infantile and Urological Sciences, Sapienza University of Rome, 00165 Rome, Italy; michele.massa@uniroma1.it; 2Department Food and Drug, University of Parma, Parco Area delle Scienze, 27/A, 43124 Parma, Italy; mirko.rivara@unipr.it (M.R.); carlotta.compari@unipr.it (C.C.); 3Department of Medicine and Surgery, University of Parma, Via Volturno, 39, 43125 Parma, Italy; thelma.deaguiarpertinhez@unipr.it; 4Department of Veterinary Sciences, University of Parma, Via del Taglio, 10, 43126 Parma, Italy; gaetano.donofrio@unipr.it (G.D.); valentina.franceschi@unipr.it (V.F.); 5Department of Mathematical, Physical and Computer Sciences, University of Parma, Parco Area delle Scienze 7/a, 43124 Parma, Italy; luigi.cristofolini@unipr.it (L.C.); davide.orsi@unipr.it (D.O.)

**Keywords:** heterocyclic *gemini* cationic surfactants, non-viral vectors, gene delivery, partially fluorinated *gemini* surfactants, atomic force microscopy on DNA, DNA-surfactant interaction, DNA circular dichroism spectra, ellipsometry

## Abstract

The development of very efficient and safe non-viral vectors, constituted mainly by cationic lipids bearing multiple charges, is a landmark for in vivo gene-based medicine. To understand the effect of the hydrophobic chain’s length, we here report the synthesis, and the chemico-physical and biological characterization, of a new term of the homologous series of hydrogenated *gemini* bispyridinium surfactants, the 1,1′-bis-dodecyl-2,2′-hexane-1,6-diyl-bispyridinium chloride (GP12_6). Moreover, we have collected and compared the thermodynamic micellization parameters (cmc, changes in enthalpy, free energy, and entropy of micellization) obtained by isothermal titration calorimetry (ITC) experiments for hydrogenated surfactants GP12_6 and GP16_6, and for the partially fluorinated ones, FGP*n* (where *n* is the spacer length). The data obtained for GP12_6 by EMSA, MTT, transient transfection assays, and AFM imaging show that in this class of compounds, the gene delivery ability strictly depends on the spacer length but barely on the hydrophobic tail length. CD spectra have been shown to be a useful tool to verify the formation of lipoplexes due to the presence of a “tail” in the 288–320 nm region attributed to a chiroptical feature named ψ-phase. Ellipsometric measurements suggest that FGP6 and FGP8 (showing a very interesting gene delivery activity, when formulated with DOPE) act in a very similar way, and dissimilar from FGP4, exactly as in the case of transfection, and confirm the hypothesis suggested by previously obtained thermodynamic data about the requirement of a proper length of the spacer to allow the molecule to form a sort of molecular tong able to intercalate DNA.

## 1. Introduction

The relevant practical interest of *gemini* surfactants, i.e., surfactants consisting of at least two identical hydrophobic chains and two polar head groups covalently bound together by a spacer, is due to their enhanced surface properties in respect to the monomeric counterparts [1,2,3,4,5]. Taking advantage of these properties, their application in biomedical and pharmaceutical fields as new drug delivery systems and as non-viral vectors for gene delivery has sparked great interest. The internalization of genetic material inside mammalian cells is only possible using vectors, both viral and non-viral. Non-viral vectors are constituted by cationic lipids able to bind and compact the genetic material into soft nanoparticles of tuneable size, so that it is protected against the action of endo- and exonucleases present in physiological fluids. Very recently, the spreading worldwide of the SARS-CoV-2 virus has focused scientific and economic efforts on searching for vaccines to fight the virus. DNA- and RNA-based vaccines, able to express the spike protein when internalized in cells, are now the most-used around the world [6,7]. Non-viral vectors are in many cases preferred, particularly when RNA-based, which require only to overcome the external cell membrane, instead of DNA-based which, needing to reach the nucleus for expression, must be introduced inside the cells. In fact, viral vectors, notwithstanding their superior efficiency, could give rise to adverse or immunogenic reactions or replications, limiting their use. Biomedical and biopharmaceutical applications of cationic *gemini* amphiphiles as gene delivery vectors, together with their chemico-physical and aggregation properties and the effect on the efficiency of transfection of their chemical structure have been extensively reviewed [5,8,9]. Non-viral vectors, obtained by synthetic route, have the advantage of being more reproducible than viral ones, and give us the possibility to understand the effect of the various moieties constituting the molecule and, eventually, to add to the vector a chemical group that can be recognized by the target cell. For many years our research was devoted from a synthetic, thermodynamic, and biomedical point of view to the study of new cationic *gemini* surfactants, having as polar head two pyridinium moieties, with the aim to find out structure–activity relationships useful for the optimization of their gene delivery ability, through the modulation of the lengths of the hydrophobic chains and spacer, counterion, and other structural modifications [10,11,12,13,14,15,16,17,18]. Taking advantage of this opportunity, we have considered not only hydrogenated *gemini* surfactants, but also the corresponding partially fluorinated (otherwise called “hybrid surfactants”) with the idea of obtaining non-viral vectors able to protect the genetic material also in those biological fluids containing endogenous hydrogenated interfering surfactants, as pulmonary surfactants or bile salts, able to destroy the lipoplexes before they enter inside the diseased cells [19,20,21]. Fluorinated surfactants are at the same time highly hydrophobic and lipophobic due to the structure of the fluorine atoms having larger van der Waals radii and lower polarizability than the hydrogen atoms, and do not form mixed micelles with hydrogenated surfactants [22,23,24]. Fluorination of cationic lipids has been proposed, among others, in the treatment of cystic fibrosis and cystic fibrosis-associated diseases [20]. Initially, we have studied the solution thermodynamics of new compounds having spacers of three, four, eight, and twelve carbon atoms, using direct methods and paying attention to the trends of partial molar volumes, compressibilities, and enthalpies vs. concentration [10,11,12,13,14,15,16,17,18]. Particularly, the measurement of their solution enthalpies gives us the key for understanding their transfection activity. Between the hydrogenated compounds, those with spacers formed by four carbon atoms shows unexpected enthalpic properties vs. concentration, which was explained by a conformation change of the molecule in solution due to stacking interactions between the two pyridinium rings. In this way, the molecule behaves as a molecular tong, able to grip the DNA bases, giving rise to a transfection activity comparable to that of the commercial reagent. If the hydrophobic chain is modified by partial fluorination, a greater length of the spacer is needed to fold the molecule. The comparison with the hydrogenated analogues reveals a greater ability of the partially fluorinated compounds to compact DNA, but only in presence of DOPE. In a recent paper [18], we tried to understand if the transfection ability is limited to a given value of spacer length, or spans over a range of values, by synthesizing the compounds, both hydrogenated and partially fluorinated, with spacer constituted by six methylene groups, namely, 1,1′-bis-hexadecyl-2,2′-hexane-1,6-diyl-bis (pyridinium) chloride (GP16_6) and 1,1′-bis(3,3,4,4,5,5,6,6,7,7,8,8,8-tridecafluorooctyl)-2,2′-hexane-1,6-diyl-bis (pyridinium) chloride (FGP6) to fill the gap between active and non-active compounds. We have outlined the completely different behaviour between hydrogenated and fluorinated compounds: in the case of hydrogenated compounds, only spacer four can deliver genes inside the cell, independently of the use of DOPE. On the contrary, fluorinated compounds having a spacer with six or eight methylene groups give rise to a very high transfection activity only in the presence of DOPE. This result indicates that the spacer length appropriate for activity falls in a broader, but always limited, range of values (spacer 12 is inactive for both classes of compounds). It is known that the critical micelle concentrations (cmc) of the fluorinated surfactants, determined by the balance of the hydrophobicity and hydrophilicity of the molecule, are close to those of ordinary surfactants whose hydrocarbon chain lengths are about 1.5 times longer [3]. This means that the -CF_2_ group is 1.5 times more hydrophobic than the -CH_2_ group. To compare in a correct way the behaviour of FGP6 with the corresponding hydrogenated surfactants having a similar hydrophobicity of the alkyl chain, we synthesized the new compound 1,1′-bis-dodecyl-2,2′-hexane-1,6-diyl-bis (pyridinium) chloride, (GP12_6), similar to GP16_6, but with an alkyl chain 12 carbon atoms long. The present paper reports the chemico-physical and tensidic properties of the compounds with spacer 6, together with the synthesis and the biological characterization of the new compound 1,1′-bis-dodecyl-2,2′-hexane-1,6-diyl-bis (pyridinium) chloride (GP12_6), to evaluate the effect of the hydrophobic chain length.

## 2. Results and Discussion

### 2.1. Micelle Formation Thermodynamics

Isothermal titration calorimetry (ITC) constitutes a particularly useful technique for studying thermodynamics of micelle formation. In fact, from the same titration experiment the thermodynamic parameters of micelle formation can be extracted, namely the critical micelle concentration (cmc), in relation to the change in free energy of micellization (Δ*G*_mic_), and the enthalpy of micelle formation (Δ*H*_mic_). The change in micellization entropy (Δ*S*_mic_) is obtained by difference. By adding the surfactant solution at a concentration of least ten time the cmc in the calorimetric cell containing water, micelles are destroyed, and the heat observed is mainly relative to the process of micelle disruption. Above the cmc region, the heat due to the process of micelle dilution is registered. The micellization parameters are obtained by extrapolating at the cmc the trends of dilution enthalpies before and after cmc, i.e., by applying a pseudo-phase transition model, in which the aggregation process is considered as a phase transition, taking place at equilibrium. In the literature [25], two different ways of obtaining the cmc values from ITC measurements are reported: (1) as the concentration of the crossing point between extrapolated initial and linear ascent lines where 1% of the surfactants is in micellar form; (2) as the inflection point in the titration curve. Both methods give about the same value when the surfactant micelles have a great aggregation number, but it is not so when the transition between monomers and micelle zone is not sharp. We have chosen, in agreement with the greatest part of the data reported in the literature, to find out the cmc as the inflection point of the titration curve, notwithstanding the not-straightforward detection. In our previous studies of solution thermodynamics of this class of *gemini* surfactants, we have measured the dilution enthalpies using a batch flow calorimeter diluting 1:1 with water the surfactant solutions prepared at different concentrations, and expressing the experimental data in terms of apparent and partial molar quantities of the solute referred to the infinite dilution as reference state [12,13,26]. When the curves of the dilution enthalpies vs. concentration obtained from calorimetric titration experiments are directly used to extract the thermodynamic micellization parameters, the reference state is the concentration of the titrating solution; this explains why, using different starting concentrations, different micellization enthalpies are obtained [27].

Therefore, ITC experiments, both for partially fluorinated and hydrogenated compounds, were performed by diluting a solution of 20 mM of surfactants so that the comparisons between the compounds here studied are meaningful, being referred to the same standard state. Figure 1 shows the ITC output for the compounds under investigation and the corresponding plots of the dilution enthalpies vs. the surfactant concentration in the calorimetric cell, starting from a titrating solution 20 mM, i.e., the trends of apparent molar enthalpies with reference state of 20 mM. From these curves, the cmc and the micelle formation enthalpy, Δ*H*_mic,_ are obtained using a pseudo-phase transition model. The micellization free energy, Δ*G*_mic,_ is calculated from equation:Δ*G*_mic_ = R*T* ln cmc
where the cmc is expressed as mole fraction, without considering the degree of counterion binding, β [28]. Moreover,
Δ*S*_mic_ = (Δ*G*_mic_ − Δ*H*_mic_)/*T*

The cmc values and the thermodynamic parameters for the micellization process so obtained are reported in Table 1.

Data shown in Table 1 confirm the similarity in hydrophobicity of FPG6 and GP12_6, allowing us to attribute the different behaviour in gene delivery ability exclusively to the fluorinated moieties. Moreover, we recall that thermodynamic properties of hydrogenated and fluorinated surfactants in solution, strictly related to the ability to compact DNA, depend both on the difference in size between fluorine and hydrogen atoms (van der Waals radii, F = 1.35 and H = 1.2 Å) and on the difference in electronegativity (Pauling scale, F = 4.0 and H = 2.1) [24].

### 2.2. Biological Assays of GP12_6

As outlined before, we have found that the partial substitution of the hydrogenated chains by shorter fluorinated ones gives rise to a very interesting gene delivery ability, as for FGP6 [18] and FGP8 [17]. Because it is known that the -CF_2_ group is 1.5 times more hydrophobic than the -CH_2_ group [29], confirmed by the data reported in Table 1, for a correct comparison of the behaviour of FGP6 with the corresponding hydrogenated surfactant, it makes sense to report here the biological data relative to the newly synthesized GP12_6.

As already conducted for the previously synthesized terms of this series [17,18], we started by testing the cytotoxicity of GP12_6 on RD4 and on A549 cells by an MTT proliferation assay. Results are shown in Figure 2 in comparison with GP16_6 [18].

By analogy with similar compounds previously studied [18], we have tested the newly synthesized compound on the human rhabdomyosarcoma RD4 cell line and on adenocarcinoma human alveolar basal epithelial cells A549, used as models for the study of lung cancer and for the development of drug therapies against it. Because this information is needed for planning transfection experiments, we performed our cytotoxicity assays also in presence of DOPE, which is essential in transfection experiments. Results in Figure 2 clearly show that the cytotoxicity of GP12_6 is very low on both kinds of cell lines, without significative differences with and without DOPE and that the cytotoxicity increases with increasing the hydrophobic chain length. The surfactant with longer tails interacts more strongly with the cellular membrane, and their complexes with the DNA could penetrate inside the cell more easily. However, their action is hindered by the increased cytotoxicity for gene delivery purposes. In fact, it is reported that the biological activity of cationic surfactants increases with the chain length up to a critical point [30]. In the case of the homologous series of alkanediyl-α,ω-bis-(dimethylalkylammonium bromide), the member with 2 alkyl chains of 16 carbon atoms is generally the most biologically active [31].

Information about the ability of the compounds under investigation to interact with DNA is obtained with EMSA experiments. Figure 3 upper shows that GP12_6 does not shift the DNA, not even at the highest concentration used, whereas GP16_ shifts the DNA at a concentration of 200 μM.

EMSA results are confirmed by morphological study by AFM in tapping mode (Figure 3 lower), using circular DNA, as described in the experimental section. The plasmid DNA alone deposited onto freshly cleaved mica was imaged as control. The images in the presence of the cationic surfactants were taken at a N/P ratio = 0.5, corresponding to the highest concentration of surfactant used in EMSA experiments. As expected, AFM images confirm that the hydrogenated surfactants with a spacer six methylene long cannot compact enough DNA in nanoparticles. In the presence of GP12_6, the AFM image is unmodified in respect to DNA alone, whereas GP16_6 starts to give rise to very loose aggregates.

The ability of GP12_6 to deliver DNA inside cells was studied with a transient transfection assay. Results on RD4 and A549 cells as a function of concentration are reported in Appendix A. The experiments were conducted with the surfactant alone and with surfactant:DOPE = 1:2, because previous experiments [17,18] have shown how the presence of DOPE can greatly enhance transfection when fluorinated surfactants are used to form lipoplexes. According to EMSA and AFM data, GP12_6 cannot act as non-viral vector both in presence and in the absence of DOPE, as GP16_6.

### 2.3. CD Spectroscopy

For better understanding the ability of the compounds under study to interact with DNA, we have undergone CD measurements, which are very sensitive to structural changes in biological molecules.

The pEGFP-C1 plasmid DNA far UV spectrum (220–320 nm) shows the contribution of right-handed A- and B-forms: where the positive band is due to base stacking, and the negative band is due to the polynucleotide right-handed helicity [32] (Figure 4). The A-form is characterized by an intense positive band around 270 nm and a weak negative band near 235 nm, and the B-form by a positive band around 280 nm and a negative at 240 nm.

Changes in the polynucleotide CD secondary structure upon surfactant addition result from their interactions. We kept the concentration of pEGFP-C1 plasmid DNA constant, and the effect of variable amounts of the 6-spacer surfactants GP16-6, GP12-6 and FGP8-6 (molar ratio 1:1, 1:2 and 1:4) were evaluated.

Upon the addition of surfactant to the DNA solution, different spectra were observed depending on the surfactant type, the length of the chain, hydrophobicity, and the molar ratio.

A weak interaction was observed adding GP12-6 at all molar ratios, which produced a shift and decrease in the positive band at 280 nm, attributed to the B-form (Figure 4).

GP16-6 affects the polynucleotide base stacking primarily. The spectra (Figure 5) show the resolution of the convoluted original positive band of the two positive bands characteristic of the A- (270 nm) and B- (283 nm) forms, respectively. We interpret this change in the CD spectrum as the result of a decrease in the A-form and an increase in the B-form.

Differently, the interaction with FGP8-6 is more complex and depends on the molar ratio (Figure 6). At a 1:1 molar ratio, the A-form decreases, as suggested by the reduction of its characteristic bands. At a molar ratio of 1:2, there is an evident shift of the negative band to the B-form and the convolution of the A- and B-form gives rise to positive bands. At a higher molar ratio (1:4) a clear increase in the B-form (278 nm) is observed. The appearance of a positive band at 288 nm wavelength (at 1:2 and 1:4 molar ratio) indicates DNA compaction [33]. The “tail” in the 288–320 nm region depends on the presence of FGP6 and it is attributed to a chiroptical feature named ψ-phase. Such a phase is a consequence of the DNA structural transition from the B-form induced by the surfactant and reflects the supramolecular structure of the complex [34]. The plasmid is tightly packed together, forming a highly condensed structure [35,36] as shown in the AFM image reported in ref. [18]. In Appendix A the deconvolutions of CD spectra (Appendix A) and the table (Appendix A) with the bands obtained using a gaussian model with Origin-Pro 2021 are reported for clarity.

### 2.4. Ellipsometry and Surface Pressure

Because the partially fluorinated surfactants are the more interesting surfactants for gene delivery purposes, we focalized our attention on the study of FGP4 (lacking transfection ability), and FGP6 and FGP8 (with a very noteworthy transfection ability) as a function of the spacer length. Initially, we studied the kinetics of surface layer formation to find out how much time these compounds take to reach equilibrium. Kinetics evolved over several hours (see Appendix A). FGP4 100 μM reached equilibrium after about 6 h, while equilibrium of FGP4 50 μM was not reached even after 15 h. Due to this slow tendency to reach equilibrium, we decided to make the lowering of surface tension and ellipsometry measurements after two hours, a range of time that is compatible with their persistence in the body, and at the same concentrations used for the transfection and cytotoxicity assays, i.e., 2.5 μM, 5 μM, 10 μM and 20 μM, concentrations well below the cmc. Therefore, all experiments were performed at a partial coverage of the surface due to the low concentrations we are interested in, and the time chosen for the measurement, relevant for the transfection experiment but not enough to reach equilibrium.

Under the above conditions, by ellipsometry at the incidence angle Φ = 55°, we measured the optical thickness of the film that is spontaneously formed at the initially clean air-water interface. We report in Figure 7 the variation of phase angle ∆, with respect to the value of the bare air/water interface ∆_0_, as a function of time and of surfactant bulk concentration. At the highest concentrations of surfactant studied, this amounts to roughly the same value for all the molecules, of the order of a variation of $\Delta -\Delta_{0}\approx -1.5^{o}$.

The small film thickness ensures we are within the Drude approximation. In this case, the variation of ∆ is proportional to the film thickness multiplied by a factor proportional to the difference between the refractive index of the film and that of the subphase (*n* = 1.33 for water as in the present case). It is, therefore, crucial to know the film’s refractive index. Unfortunately, there are no data in the literature as these are newly synthesized molecules, and it is difficult to predict it a priori.

A first estimate can be drawn based on the molecular structure: FGPs molecules contain two types of units, fluorinated and hydrogenated. Fluorinated alkyl chains, such as perfluorohexanoic acid (CAS 307-24-4) have *n* = 1.301, pyridine has *n* = 1.51, while alkyl chains have typically n≃1.40 at the relevant wavelength. We can attempt some rough volumetric considerations, assuming the polar head comprising of pyridine and alkyl spacer with thickness ∼ 3–4 A, and average refractive index 1.46–1.47 depending on alkyl spacer length, while the *gemini* partially fluorinated chains, common to all the molecules, account for ∼ 2 + 6 A of thickness and refractive index *n* = 1.40 and *n* = 1.30 for the protonated and fluorinated chains, respectively.

Basing on such very broad considerations, we can estimate the molecular refractive index to be not too far from *n* = 1.37–1.39. With these values, film thickness results to be of the order of 15–20 A, thus compatible with the formation of a monolayer at the interface. A more accurate estimate of film thickness could be made by using X-ray reflectometry, while neutron reflectometry could provide even more detailed insights into the film structure [37].

The relative weight of the fluorinated part (small refractive index *n*) in respect to the hydrogenated part (big refractive index *n*) is larger in FGP4 than in FGP6 and FGP8. This suggests that the refractive indexes of FGP4 is the smallest and closest to water, followed by FGP6, while FGP8 is the largest. This explains, in part, why FGP4 has the smallest variation of ∆.

At lower concentrations, however, the different molecules exhibit a very different behaviour, with FGP6 and FGP8 displaying a lower variation of ∆ than FGP4. This cannot be accounted for by the previous argument on the contribution of fluorinated chains to the refractive index (which would predict a smaller variation for FGP4) and can only be rationalized assuming that the films formed by different molecules have different thicknesses: in films formed by FGP6 and FGP8 at low packing, thanks to the longer alkyl spacer, the fluorinated chains may lay oblique and close to the water surface, while on the contrary in FGP4 even at low coverage the molecules are forced to stand more vertical due to the shorter alkyl spacer.

Coherently with this explanation, we also find that in this regime of lower concentration, the surface pressure ∏, i.e., the reduction of interfacial tension from its value for the bare air-water interface γ=72.6 mN/m, is greater for FGP6 and FGP8 than for FGP4, especially at the lower concentrations (Figure 8).

In the same Figure, we also show the limiting pressures that would be reached by the same surfactants at much higher concentrations, above their respective CMC values.

In the same Figure, we also report previously published data for FGP4 and FGP8 at concentration up to and above cmc, adapted from ref. [16]. In summary, it is also important to notice that FGP6 and FGP8 act in a very similar way, and dissimilar from FGP4, exactly as in the case of transfection. This suggests the presence of a dichotomy: either the spacer is or is not long enough for the flexibility required by the molecule to accommodate at the interface in the present case, or around a DNA fragment in the case of interest for transfection.

## 3. Materials and Methods

### 3.1. Compounds

All the compounds under study with six methylene spacers, hydrogenated GP*m*_6, where *m* il the hydrophobic tail length (*m* = 12 and 16), and partially fluorinated FGP*n* (where *n* is the spacer length, *n* = 4, 6, 8) with hydrophobic tails constituted by two -(CH_2_)_2_-(CF_2_)_5_-CF_3_ groups, were prepared by us. The synthesis of GP16_6 and FGP6 are reported in ref. [18], that of FGP4 and FGP8 in ref. [16]. We synthesized for the first time GP12_6, having a hydrophobicity comparable to FGP6 (see text), and its synthesis is here reported. The structure of GP12_6 and of the compounds with spacer six methylene long under study is shown in Figure 9.

### 3.2. Chemical Synthesis

We synthesized the new compound 1,1′-bis-dodecyl-2,2′-hexane-1,6-diyl-bispyridinium chloride (GP12_6), following and adapting, when necessary, the procedures previously reported [11,16].

Synthesis of 1,1′-bis-dodecyl-2,2′-hexane-1,6-diyl-bispyridinium chloride: 1,6-bis(2-pyridyl)hexane (1.62 g, 0.007 mol) was dissolved in 10 mL of DMF. The temperature was raised to 150 °C and dodecyl chloride (14.3 g, 0.07 mol) was slowly added to the solution. The reaction mixture was stirred for 18 h and the DMF was evaporated under reduced pressure. The residue was suspended in diethyl ether and crystallized two times from acetonitrile/toluene giving yellow crystals. The synthesis is reported in Figure 1. Yield 2.64 g (58%); ^1^H-NMR (400 MHz DMSO-*d*_6_): δ 0.86 (t, *J* = 7.9 Hz, 6H); δ 1.23 br s, 40H); δ 1.72 (m, 4H); δ 1.84 (m, 4H); δ 3.09 (m, 4H); δ 4.56 (t, *J* = 8 Hz, 4H); δ 8.00 (m, 4H); δ 8.50 (t, *J* = 8 Hz, 2H); δ 9.03 (*d*, *J* = 8 Hz, 2H). ^13^C NMR (100 MHz, DMSO-*d*_6_): δ 158.2, 149.3, 136.7, 123.6, 121.3, 55.8, 44.2, 34.4, 32.7, 31.7, 29.5, 28.8, 26.7, 24.4, 22.6, 14.4, 13.8. FT-IR: ν = 3479, 3432, 2912, 2849, 1638, 1470, 1012, 933, 715 cm^–1^. MS-ESI: *m*/*z* = 614 [M-Cl]. Anal. Calcd. For C_40_H_70_Cl_2_N_2_ (649.91): C 73.92, H 10.86, N 4.31. Found C 74.09, H 10.72, N 4.12.

The NMR spectra were recorded using a Bruker 400 Avance (Billerica, MA, USA) spectrometer.

^1^H NMR Spectra (400 MHz) chemical shifts (d scale) are reported in parts per million (ppm) and reported in order: multiplicity and number of protons; signals were characterized as s (singlet), d (doublet), t (triplet), m (multiplet), br s (broad signal).

^13^C NMR (100 MHz); chemical shifts (d scale) are reported in parts per million (ppm).

IR spectra were recorded using an Agilent Technologies Cary 630 FTIR Spectrometer in the region 700–4000 without KBr support.

Mass spectra were recorded using an Applied Biosystem/MDS SCIEX API-150 EX instrument (Waltham, MA, USA).

The new compounds were analyzed on a ThermoQuest (Rodano, Italy) FlashEA 1112 Elemental Analyzer, for C, H, N. The percentages recorded were within 0.4% of the theoretical values.

The NMR, IR and mass spectra can be found in Appendix A (Appendix A).

### 3.3. ITC Measurements

ITC measurements of the dilution enthalpies of the surfactants under study were carried out on a MicroCal PEAQ-ITC (Malvern) at 25 °C. Surfactant solution was injected (first injection of 0.4 μL, followed by 18 injections of 2 μL each or 25 injections of 1.5 μL each) into a 200 μL reaction cell filled with doubly distilled and degassed water (as well as the reference cell) using a 40 μL automatized syringe with an interval of 150 or 120 s between two successive injections. A continuous stirring at 150 rpm was maintained throughout the experiments. The surfactant solutions used as titrants were prepared by weight using freshly boiled and degassed bi-distilled water at the concentrations of 10 and 20 mM. Data analysis was performed using MicroCal PEAQ-ITC Analysis Software (version 1.41, Malvern Panalytical, Malvern, UK).

### 3.4. Biological Assays

The biological assays of the new compound were carried out as described in ref. [18] to which we refer for experimental details. The human rhabdomyosarcoma cell line RD-4 (ATCC^®^ CCL-136™) and the human pulmonary adenocarcinoma cell line A549 (ATCC^®^ CCL-185™), cultured as described in ref. [18] were used in the experiments. Toxicity was evaluated with the MTT proliferation assay and the statistical differences among treatments were calculated using Student’s test and multi-factorial ANOVA. Electrophoresis Mobility Shift Assay (EMSA) was used to evaluate the interaction between the cationic surfactant and the pEGFP-C1 plasmid, expressing the fluorescent green protein, prepared and stored as in ref. [18]. With a transient transfection assay, the gene delivery ability to the above cell lines of the compound under study was evaluated. Lipoplex formulations were prepared with the surfactant alone and by adding 1,2-dioleyl-sn-glycero-3-phosphoethanolamine (DOPE; SIGMA-Aldrich, St. Louis, CA, USA) to the plasmid–surfactant mixture at a surfactant:DOPE molar ratio of 1:2. Transfected cells were observed under a fluorescence microscope for EGFP expression.

### 3.5. Sample Preparation and AFM Imaging

A 20 μL droplet of a solution 0.1 nM of plasmid DNA in deposition buffer (4 mM Hepes, 10 mM NaCl, 2 mM MgCl_2_, pH = 7.4), either in the presence or in the absence of the cationic lipid, was deposited onto freshly cleaved ruby mica (Ted Pella, Redding, CA, USA) after 5 min incubation at room temperature. The same N/P ratio, i.e., the ratio between the negative charges of the phosphate groups of DNA and the positive charges carried by the *gemini* pyridinium surfactant, as in transient transfection experiments, was used [18]. The mica disk was rinsed with Milli-Q water and dried with a weak nitrogen stream.

AFM imaging was carried out on the dried sample with a Park XE-100 microscope, operating in tapping mode, using commercial diving board silicon cantilevers (NSC-15 Micromash Corp., Sofia, Bulgaria). The software XEI (Park Systems, Suwon, Korea) was used for optimizing the images obtained by Park XE-100.

### 3.6. Circular Dichroism Spectroscopy

Circular dichroism (CD) experiments were performed using a Jasco 715 spectropolarimeter (JASCO International Co. Ltd., Tokyo, Japan), coupled with a Peltier PTC-348WI system for temperature control, set at 25 °C. CD spectra were the average of 4 scans recorded in the 220–320 nm range, using a 1 mm path length quartz cuvette, a bandwidth of 1 nm, data pitch of 0.5 nm, and a response time of 8 s. 

Spectra were performed at a fixed pEGFP-C1 plasmid DNA concentration (1.36 × 10^−3^ M, with molarity expressed in terms of base pairs, bp) and in varying amounts of *gemini* surfactants to obtain the molar ratio 1:1, 1:2, 1:4 in 5 mM HEPES buffer, pH 7.0. 

Following baseline correction, the measured ellipticity, θ (mdeg), was converted to the molar mean ellipticity [θ] (deg·cm^2^·dmol^−1^), using [θ] = θ/10 cl, where θ is ellipticity, c is the DNA molar concentration, and l is the optical path length in centimeters. 

CD spectra deconvolution was performed using a gaussian model with Origin-Pro 2021.

### 3.7. Surface Tension Measurements

Surface tension measurements were performed using the Wilhelmy plate method, bringing a filter paper plate into contact with the surface of the liquid. The Wilhelmy plate was 0.75 square centimeters in area (0.5·× 1.5), attached to a balance with a thin metal wire. The force F on the plate due to wetting is measured with a NIMA PS4 tensiometer equipped with thermostatic support, to keep the temperature constant and avoid evaporation of water, and a trough in Teflon. The surface tension (*γ*) is calculated using the Wilhelmy equation:$$\gamma =\frac{F}{l\;\cos\left(\theta \right)}$$
where *l* is the wetted perimeter of the Wilhelmy plate and θ is the contact angle between the liquid phase and the plate. In all the measurements reported here it was found θ = 0, i.e., complete wetting conditions [38].

### 3.8. Ellipsometry

Ellipsometry measurements were performed using a multipurpose apparatus (Multiskop Optrel, Berlin, Germany) with a single wavelength of 632.8 nm. This technique was employed to evaluate the thickness of the surface layer, always in the Drude approximation [39]. The optical model adopted was that described in detail in ref. [40]. Film thickness was probed by the variation of the ellipsometric phase-angle δ∆ which, in the Drude approximation, ref. [39], is linearly proportional to film thickness via a scaling factor depending on the incidence angle and refractive indexes. This specific apparatus provides an average thickness value calculated on an area of a few square millimetres.

## 4. Conclusions

Gene-based medicine is a clinical reality initially thought to fight against acquired or inherited genetic diseases and, after the pandemic spreading of the SARS-CoV-2 virus, well-known worldwide due to the use of vaccines based on viral DNA or mRNA. A great discussion is still open among scientists about the efficacy and the safety of this medical treatment, and research in this field is increasing more and more. Regardless of the kind of genetic material to be introduced inside the cells, it must be delivered by means of vectors, and the development of very efficient and safe non-viral vectors, avoiding immunogenic reaction and viral replication, is a landmark for in vivo gene-based medicine. We have devoted many years of our research to the synthesis, and chemico-physical and biological characterization, of *gemini* bis-pyridinium surfactants, both hydrogenated and partially fluorinated, and the results obtained confirm that pyridinium *gemini* surfactants, particularly those partially fluorinated, could be valuable tools for gene delivery purposes, but their performance highly depends on the spacer length and is strictly related to their structure in solution. To find out structure–activity relationships necessary to optimize their performance, we are collecting their chemico-physical properties in relation to their biological behaviour. The synthesis of the new hydrogenated compound having the spacer six methylene long and the hydrophobic tails of twelve carbon atoms each shows that the gene delivery ability of the hydrogenated compounds depends on the spacer length but barely on the hydrophobic tail length. Moreover, the physico-chemical data, particularly cmc and enthalpy of micellization, confirm that GP12_6 has a hydrophobicity very similar to FGP6, making their comparison correct. CD spectra have been shown to be useful tools to verify the formation of lipoplexes. In fact, under lipoplex formation, the CD spectra show a “tail” in the 288–320 nm region dependent on the presence of FGP6 and attributed to a chiroptical feature named ψ-phase. Such a phase is a consequence of the DNA structural transition from the B-form induced by the surfactant and reflects the supramolecular structure of the complex.

We have previously shown that the fluorinated compounds with spacer formed by 6 (FGP6) and 8 (FGP8) carbon atoms gives rise to a very interesting gene delivery activity, superior to that of the commercial reagent, when formulated with DOPE. Ellipsometric measurements suggest that FGP6 and FGP8 act in a very similar way, and dissimilar from FGP4, exactly as in the case of transfection, and confirm the hypothesis, suggested by thermodynamic data, about the requirement of a right length of the spacer for allowing the molecule to form a sort of molecular tong able to intercalate DNA. It remains to verify the effect of the hydrophobic chain length on fluorinated compounds, to optimize their efficiency.

## Data Availability

All experimental data are reported in the paper and in Appendix A.

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
