# Peer review of "Chemico-Physical Properties of Some 1,1′-Bis-alkyl-2,2′-hexane-1,6-diyl-bispyridinium Chlorides Hydrogenated and Partially Fluorinated for Gene Delivery"

_molecules, 2023, doi:10.3390/molecules28083585_

Round 1
Reviewer 1 Report
Manuscript by Massa M et al. brings new knowledge in the field of amphiphilic compounds and gene delivery. there is a strong assumption that it will be interesting to many readers. The manuscript is written at a good level, I have a few comments about it:
1. Chemical names o fcompounds should be uniform (E.g. line 22 and 94/95). I recommend replacing „hexamethylene“ with „hexane-1,6-diyl“
2. Please check the citation (E.g. line 122)
3. Line 135 Why „different steps of the synthesis“?
4. Interaction constants in NMR spectra at t and d are missing.
5. Line 197 Down index for 2 at MgCl2
6. I recommend confirming the cmc values also using Wilhelmy plate methods.
7. I recommend draw structure of all atudied compounds.
Reviewer 2 Report
In this manuscript, Fisicaro and coworkers synthesized a series of hydrogenated gemini bispyridinium surfactants and investigated their chemical, physical and biological properties. Analyses included EMSA, MTT, AFM imaging, etc. Systematic studies for GP12_6, GP16_6, and FGPn series revealed the gene delivery properties are dependent on the spacer length instead of tail length. The study strongly suggested the potential of using these compounds in the gene delivery area. Overall, the manuscript is well-written, and the experiments are well-designed. The reviewer would suggest the publication of the manuscript in Molecules after some minor revisions:
1. It would be beneficial to show the chemical structures of all the surfactants studied in the manuscript, which would make it easier for the readers.
2. The manuscript’s title doesn’t seem consistent for different files.
3. For scheme 1, compound 2 appears to be GP12_6. The authors may want to change the number to minimize confusion. Also, it is recommended to put yields on the reaction scheme. 4. The punctuation for decimal places needs to be consistent throughout the manuscript. For example, in the heading for Fig 2, the decimal signs differ in (f) and (g).
